# Dual-Wavelength Lasing with Orthogonal Circular Polarizations Generated in a Single Layer of a Polymer–Cholesteric Liquid Crystal Superstructure

**DOI:** 10.3390/polym15051226

**Published:** 2023-02-28

**Authors:** Donghao Yang, Marouen Chemingui, Yu Wang, Xinzheng Zhang, Irena Drevensek-Olenik, Faheem Hassan, Qiang Wu, Yigang Li, Lotfi Saadaoui, Jingjun Xu

**Affiliations:** 1The MOE Key Laboratory of Weak-Light Nonlinear Photonics and International Sino-Slovenian Joint Research Center on Liquid Crystal Photonics, TEDA Institute of Applied Physics, School of Physics, Nankai University, Tianjin 300457, China; 2LR99ES16 Laboratoire de Physique de la Matière Molle et de la Modélisation Electromagnétique, Faculté des Sciences de Tunis, Département de Physique, Université de Tunis El Manar, Tunis 2092, Tunisia; 3Collaborative Innovation Center of Extreme Optics, Shanxi University, Taiyuan 030006, China; 4Faculty of Mathematics and Physics, University of Ljubljana, Department of Complex Matter, J. Stefan Institute, SI-1000 Ljubljana, Slovenia

**Keywords:** cholesteric liquid crystals, polymer–liquid crystal, polymer composites, filler, dual-wavelength lasing, orthogonal circular polarizations

## Abstract

We investigate the laser emission from a polymer–cholesteric liquid crystal superstructure with coexisting opposite chiralities fabricated by refilling a right-handed polymeric scaffold with a left-handed cholesteric liquid crystalline material. The superstructure exhibits two photonic band gaps corresponding to the right- and left-circularly polarized light. By adding a suitable dye, dual-wavelength lasing with orthogonal circular polarizations is realized in this single-layer structure. The wavelength of the left-circularly polarized laser emission is thermally tunable, while the wavelength of the right-circularly polarized emission is relatively stable. Due to its relative simplicity and tunability characteristics, our design might have broad application prospects in various fields of photonics and display technology.

## 1. Introduction

Circularly polarized light provides an additional degree of freedom for photonics applications because it carries an optical spin that possesses optical rotatory power and invariability under rotations [1]. In recent years, circularly polarized laser emission has attracted a lot of attention and become a vivid topic in the field of photonics. In particular, the separate control of left-handed and right-handed circularly polarized laser emissions shows broad application potential in advanced spectroscopies [2], 3D display technologies [3], information storage and processing [4], orbital angular momentum micro-laser fabrication [5], the communication of spin information [6], and ellipsometry-based tomography [7]. In order to realize circularly polarized laser emission, the possibility of embedding a laser gain medium into a chiral environment is commonly required. Dye-doped cholesteric liquid crystals (CLCs) fulfill this requirement; therefore, they are very suitable candidates for fabricating laser devices that generate circularly polarized light [8].

It is well-known that CLCs act as one-dimensional photonic crystals because of their spontaneously formed periodic helical structure. The corresponding photonic band gap (PBG) leads to the selective reflection of circularly polarized light with the same handedness as the helix. The photon group velocity at the band edges approaches zero, which makes CLCs convenient materials to achieve laser emission [9,10,11,12,13,14,15,16]. CLC lasers have gained significant attention because of their various advantages, such as their self-assembling helical structures, large coherence area, and potential for multidirectional emission. In addition, CLCs are especially famous for their PBGs that are adjustable by external fields, such as electric fields [17], temperature [18], pressure [19], light [20], and magnetic fields [21], which make CLCs popular in industrial use and research, such as that of display and laser technologies and other photonic devices. However, the thermal orientational fluctuations of the fluidic LC molecules in CLCs can significantly distort the resonant structure and increase the energy threshold for lasing emission, which may decrease the performance of CLC lasers.

Recently, a novel kind of CLC superstructure, fabricated using UV-curing and washing out/refilling processes, has been developed to fabricate CLC photonic devices with a high stability that provide a high-quality chiral environment for either the reflection or emission of light [22]. A polymer scaffold with a helical structure was prepared by photopolymerizing photoreactive mesogens that were dispersed within a CLC material. The CLC initially defined the pitch of the structure but was then entirely removed. By refilling such a scaffold with an optically isotropic material [23], a nematic liquid crystal [24], or a CLC with the same/opposite chirality [25], various photonic devices, such as low-threshold lasers [26], wide-band reflection layers [27], and highly reflective devices [28], have been previously demonstrated. This kind of superstructure has also been used to achieve the simultaneous control of orthogonal circularly polarized light and the conjugate geometric phase [29]. Traditionally, in the field of CLC lasers, opposite-handed circularly polarized laser emission is achieved by means of CLC cell arrays [30], double-layer polymer network structures [31], and opposite-handed circularly polarized emission at a single wavelength [32]. All these concepts require relatively a high complexity of fabrication and a large volume, and they exhibit an untunable wavelength, so they cannot accommodate advanced capabilities for optical devices. Therefore, the idea to realize a simple tunable dual-wavelength laser device with opposite-handed circularly polarized laser emission is very challenging.

In this work, we demonstrate a method for preparing a hybrid structure by refilling a left-handed CLC into a right-handed polymeric scaffold that was fabricated from a right-handed polymerized CLC. Two reflection bands related to opposite-handed circularly polarized light were observed and showed a similar optical quality. With this superstructure, dual-wavelength lasing with orthogonal circular polarizations was realized experimentally. One of the emission wavelengths could be thermally tuned, and the other was relatively stable. These special optical properties make these novel CLC composite structures very promising for applications in many fields, such as tunable lasers and laser displays, and in various optical components and devices.

## 2. Materials and Methods

### 2.1. Material Preparation

In a CLC phase, the pitch of the helical structures (*p*) is inversely proportional to the product of the helix twist power and the concentration of the chiral dopant *p* = 1/(*C* × *HTP*), where *HTP* is the helix twist power, and *C* is the concentration of the chiral dopant. The reflection band satisfies the equation *λ*_l_ = *n*_e_·*p*, *λ*_s_ = *n*_o_·*p* for a positive birefringence liquid crystal, where *λ*_l_ and *λ*_s_ are the long and short-wavelength edges of the PBG, and *n*_e_ and *n*_o_ are the extraordinary and the ordinary refractive indices, respectively. The band width of the PBG satisfies the equation △*λ* = |*n*_o_ − *n*_e_|·*p*. The materials used to fabricate the superstructure are as follows: the right-handed polymer CLC (RH-PCLC) is a mixture of 48 wt% of non-reactive nematic liquid crystal E7 (Qingdao QY Liquid Crystal Co., Ltd., Qingdao, China), 25 wt% reactive nematic monomer RM257, 26.9 wt% right-handed chiral agent R811 (Qingdao QY Liquid Crystal Co., Ltd.), and 0.1 wt% photo-initiator Irgacure 651 (BASF), while the left-handed CLC (LH-CLC) is a mixture of 73 wt% of E7 and 27 wt% left-handed chiral agent S811 (Qingdao QY Liquid Crystal Co., Ltd.). In the E7 host, the helical twisting power values of R811 and S811 are both about 11.24 μm^−1^. Under UV illumination, the photo-initiator Irgacure 651 generates free radicals that trigger the chain polymerization process of RM257 so as to strengthen the polymer matrix. For lasing emission experiments, a laser dye PM597 (0.2 wt%) is added to the LH-CLC mixture, and the concentration of S811 is changed to 26.8 wt%.

### 2.2. Fabrication Process

A schematic diagram that explains the preparation of the superstructures is shown in Figure 1. In order to induce a planar surface alignment of LC molecules, a polyimide solution (PI-3010, POME) was spin-coated onto the surfaces of ITO-coated glass plates at a speed of 1200 rpm for 10 s and 3500 rpm for 30 s in sequence. Then, the glass plates were heated on a hot stage at 130 °C for 120 s and at 230 °C for 20 min in sequence and rubbed in the same direction with a piece of friction-oriented cloth. Plastic microspheres with a diameter of 10 μm mixed into an ultraviolet curing adhesive, NOA65, were used as the cell spacers. Liquid crystal cells were prepared using two rubbed ITO-coated glass plates assembled in an anti-parallel way with spacers at their corners.

The RH-PCLC mixture was filled into the cell in its isotropic phase (85 °C) with the help of capillary force. To obtain a uniform standing helix alignment (Grandjean texture), the sample was slowly cooled down from the isotropic phase to the cholesteric phase. A good alignment that was practically monodomain in nature was achieved. When the temperature of 32 °C was reached, the liquid crystal cell was exposed to UV light (HTLD-4II, Height-LED Opto-electronic Tech Co., Ltd., Shenzhen, China) at a wavelength of 365 nm (100 mW/cm^2^). To ensure uniform curing conditions, each sample was illuminated for 20 s on each side of the cell. After polymerization, the cell was immersed in acetone for 48 h to wash out the unreacted liquid crystal molecules, and then it was put on the hot stage at 80 °C for 3 h to remove the residual acetone. Finally, an LH-CLC mixture was refilled into the cell in the isotropic phase (85 °C) for 12 h with the help of capillary force. The refilling process of the scaffold samples took significantly longer than the initial filling process, which ensured the complete filling of the LC into the porous polymer scaffold.

### 2.3. Polarizing Optical Microscopy Study

To check the LC alignment in the samples at different preparation phases, we used a polarizing optical microscope to observe their visual textures. The textures were captured by using an inbuilt digital camera mounted at the top of the microscope and transferred to a computer. The temperature of the samples was controlled with the help of a hot stage (HCS 311i, Instec, Inc., Boulder, CO, USA) attached to the temperature controller (mK 1000, Instec, Inc.), having an accuracy of ±0.1 °C and a resolution limit of ±0.003 °C.

### 2.4. Optical Transmission Study

To measure the transmission spectrum of the samples during each fabrication step and consequently resolve the corresponding band gaps, the samples were illuminated by using a deuterium–halogen white light source (DH-2000-BAL, Ocean Optics, Duiven, The Netherlands) through a lens with a focal length of 85 mm. The samples were fixed on the hot stage (HCS 402, Instec Inc.). The transmitted light was collected by another lens with the same focal length and coupled to a spectrometer (HR4000CG-UV-NIR, Ocean Optics).

### 2.5. Lasing Emission Study

Figure 2 shows the experimental setup used for measuring the emission spectra of the sample. For optical excitation (pumping), the second harmonic output (wavelength 532 nm, repetition rate 1.0 Hz, pulse duration 4.0 ns) from a *Q*-switched Nd:YAG laser (SLIII-10, Continuum, Dallas, TX, USA) was used. The beam was split into two beams. The pulse energy was monitored in the reflected direction with an energy meter (LabMax-Top, Coherent Inc., Santa Clara, CA, USA). By using a convex lens with a focal length of 85 mm, the beam in the transmission direction was focused onto the sample, which was fixed on the hot stage. The diameter of the focal spot was about 150 μm. The incident beam was perpendicular to the cell due to the limited window size of the hot stage. In order to avoid the influence of the pump laser and protect the spectrometer, a 532 nm notch filter was put behind the hot stage. The light emitted from the sample was collected by using a 10× objective lens and passed through a quartz quarter-wave plate, with its fast axis perpendicular to the horizontal plane and a polarizer. Then, it was coupled to an optical fiber connected to a high-resolution spectrometer (SP2358, Princeton Instruments, Trenton, NJ, USA).

## 3. Results and Discussion

### 3.1. Texture Changes Observed by Using a Polarizing Optical Microscope

Figure 3 shows the polarized microscopic transmission images of the different samples. Figure 3a shows an image of the RH-PCLC before UV exposure (Sample A), which has a Grandjean texture with a blue color, proving that the CLC assembled into a uniform standing helix structure. There are some green micro-domains, which may be caused by the pre-polymerization from the light source of the microscope. Figure 3b is an image of the RH-PCLC after UV exposure (Sample B); the formed polymer scaffold compressed the helixes in space. Therefore, the color of the reactive area changed, and oily steaks appeared due to the misalignment of the liquid crystal molecules. After immersion in acetone, LC molecules, chiral dopant molecules, and unreacted RM257 were soaked out, so the polarized microscopic image of Sample C looks dark with some bright areas, as shown in Figure 3c, which is because the polymer network is composed of a liquid crystalline polymer, RM257. Figure 3d shows an image of Sample D, in which one can see many green and brown micro-domains, i.e., transverse inhomogeneity. The green spots have a high proportion of the RH-CLC, while the brown spots have a high proportion of the LH-CLC. The color variation between the green and brown suggests longitudinal inhomogeneity perpendicular to the cell surface. Moreover, there is a relatively large speckle with turquoise stripes, where there are much less right-handed polymer network areas, and LH-CLCs occupy the large gaps.

### 3.2. Two Photonic Band Gaps Corresponding to the Right- and Left-Circularly Polarized Light

It is known that the helix-structure memory of the polymer network is derived from the helical properties of the initial CLC; in other words, the polymer network memorizes the liquid crystalline order by imprinting the respective structure and its orientation into the network during polymerization. In Figure 4a, the transmission spectra of the sample with the RH-PCLC after UV exposure (Sample B) and after immersion in acetone (Sample C) are shown as red and black lines, respectively. As expected, the PBG appears in the visible range after polymerization via UV exposure. By immersing the RH-PCLC sample in acetone for 48 h, the unreacted LC and chiral dopant molecules were soaked out from the polymer scaffold and replaced by air. This caused a decrease in the average refractive indices of the structure, due to which the PBG was blue-shifted out of the visible range [33], as illustrated by the black line in Figure 4a. The transmission spectra of the LH-CLC in a conventional liquid crystal cell (Sample E) and the superstructure (Sample D) are shown in Figure 4b,c, respectively. The helical pitch of the LH-CLC decreases with the temperature, because the twisting power of S811 increases with the temperature [34]. As a result, the PBG of the LH-CLC can be thermally tuned. As shown in Figure 4b, the center wavelength of the PBG shifts from 681 nm at 27 °C to 622 nm at 30 °C. After refilling with the LH-CLC, the liquid crystal molecules move into the nanopores of the polymer scaffold. Thus, the average refractive indices increase, which causes the PBG of the structure corresponding to the right-handed circularly polarized (RCP) light to shift back to the visible range [33]. In the gaps far from the polymer network, the helix twisting power of the left-handed chiral agent (S-811) is much stronger than the anchoring force of the polymer scaffold; therefore, the PBG corresponding to the left-handed circularly polarized (LCP) light arises. The right-handed PBG from the superstructure (Sample D) has a significant blue shift compared to the RH-PCLC (Sample B) PBG, which is due to the incomplete reoccupation restricted by the high viscosity of the liquid crystal molecules and the low density of the air-filled nanopores in the polymer scaffold [33]. Besides this, the left-handed PBG of Sample D exhibits a slight blue shift compared to the LH-CLC in the conventional liquid crystal cell (Sample E) at the same temperature. This is because the molecular weight of the chiral dopant S-811 is higher than that of the composition of the nematic liquid crystal E7, which means a higher viscosity index of S-811 [35]. There is less S-811 filled in the nanopores of the polymer scaffold, and the increase in the concentration of S-811 in the gaps far from the polymer network decreases the pitch length. Moreover, because the helix twisting power of S-811 increases with the temperature, the center wavelength of the left-handed PBG of the superstructure experiences a blue shift from 656 nm at 27 °C to 610 nm at 30 °C. The low transmittance around 570 nm is due to the overlapping of two PBGs.

### 3.3. Dual-Wavelength Lasing with Orthogonal Circular Polarizations

In order to achieve laser emission, a fluorescent dye, PM597, was added into the refilled LH-CLC (Sample F). The emission spectra of Sample F at 28 °C and those of PM597 measured in the isotropic phase of the LH-CLC are shown in Figure 5. Two emission peaks at 560 nm (Peak A) and at 604 nm (Peak B) arise, which are located at the long-wavelength edge of the right-handed PBG and the short-wavelength edge of the left-handed PBG, respectively. The dual-wavelength emission spectra of Sample F under different pump energies at 28 °C are shown in Figure 6a. The RCP and LCP light can be converted to linearly polarized light by using a quarter-wave plate, which, in our case, was oriented along the directions of 135° and 45° with respect to the vertical axis, respectively. The direction of the resulting linearly polarized light can be distinguished by an analyzer. Based on Figure 5b,c, one can conclude that emission Peaks A and B correspond to the RCP and LCP light, respectively.

Figure 6d,e depict the integrated intensities of the RCP and LCP emission peaks and their corresponding full widths at half maximum (FWHM) for the dye-doped superstructure (Sample F) as a function of pump energy. The FWHMs of both peaks were obtained by fitting the fluorescence spectrum with a double Lorentzian function. When the pump energy is low, there is no laser emission. When the pump energy is larger than the threshold, the output intensity increases sharply with the increase in the pump energy. Meanwhile, the FWHM of each peak becomes significantly narrower. Both the threshold effect and spectral narrowing prove that these two emission peaks belong to lasing emission. The lasing threshold of Peak A (1.1 μJ) is slightly higher than that of Peak B (0.82 μJ). It is known that the transition dipole directions of dye molecules affect the polarization properties of their fluorescence. Laser oscillation at the short-wavelength band edge can be stronger than that at the long-wavelength band edge when the transition dipoles of dye molecules are perpendicular to the liquid crystal director. We measured the fluorescence angular distribution of PM597 dissolved in a planar-aligned LC cell filled with pure E7, as shown in Figure 7. From the obtained result, we calculated the order parameter of the transition dipole moment of the dye [36]:(1)$$S_{d}=\frac{r\;-\;1}{r+2}$$

Here, the dichroic ratio *r* is defined as *r* = *I*_∥_/*I*_⊥_, where *I*_∥_ and *I*_⊥_ are the fluorescence intensities parallel and perpendicular to the nematic director. Based on Figure 7, it follows that the dichroic ratio of PM597 is around 1.63, and that *S_d_* is around 0.173, which, in our opinion, is too small to have a large effect on the difference in the lasing thresholds at the long- and short-wavelength band edges. However, the visible PBG related to Peak A was formed by the refilling of liquid crystal molecules in the nanopores of the polymeric scaffold. It was also affected by the misalignment caused by polymer crosslinking during UV exposure. As a result, the uniformity and integrality of the right-handed segments of the structure were lower than those of the left-handed segments, and, consequently, the edge of the right-handed PBG was less sharp than the edge of the left-handed PBG, as shown in Figure 5. This means that the density of the photonic states at the short-wavelength band edge of the LH-CLC was higher than that at the long-wavelength band edge of the RH-CLC, resulting in the lower lasing threshold of Peak B.

### 3.4. Thermal Tuning of Dual-Wavelength Lasing

It has been demonstrated that the PBG of a CLC can be adjusted either by temperature modifications or by exposure to external fields, which enables the tuning of band-edge lasing emissions [20,37,38,39]. Sample F was fixed in the hot stage and heated from both sides to ensure temperature uniformity. When the temperature increased from 27 °C to 30 °C in steps of 0.5 °C, the LCP lasing showed a blue shift from 618 nm to 569 nm due to the blue shift of the left-handed PBG. During heating, the extraordinary refractive index of the nematic liquid crystal E7 slightly decreased [40]. According to the equation *λ*_l_ = *n*_e_·*p*, the long-wavelength edge of the right-handed PBG experienced a slight blue shift, which caused the RCP lasing to have a slight blue shift from 561 nm to 557 nm, as shown in Figure 8. It is conceivable that, if required, multi-wavelength lasing or even R + G + B lasing can be achieved by adding proper laser dyes and tuning the temperature in a suitable range.

In addition, the temperature can also affect the emitted intensity. As shown in Figure 8c, when the temperature increases from 27 °C to 29 °C, the emission intensity of Peak B rises gradually. This is because the spectral position of Peak B becomes closer to the fluorescence peak of the laser dye PM597 at 570 nm due to the blue shift of the PBG of LH-CLC, which means a higher gain in the lasing emission. However, as the temperature continues to rise, the emission intensity drops. We attribute this effect to the thermal accumulation in the structure that causes the misalignment of liquid crystal molecules and the disturbance of the pitch of the LH-CLC. This, consequently, causes a loss in the lasing emission and reduces the emission intensity of Peak B.

## 4. Conclusions

In conclusion, dual-wavelength lasing with orthogonal circular polarizations generated in a single-layer polymer-CLC superstructure was achieved by refilling the LH-CLC into the RH-helical polymer scaffold. Two PBGs corresponding to orthogonal circular polarizations appeared in this single-layer superstructure due to the combined helical twisting power of the chiral polymer scaffold and the CLC. After doping the mixture with the laser dye PM597, we realized orthogonal circular polarized lasing emissions with peak wavelengths at the long-wavelength band edge of the right-handed PBG and the short-wavelength band edge of the left-handed PBG. The threshold of the RCP lasing at 560 nm was 1.1 μJ, and that of the LCP lasing at 604 nm was 0.82 μJ at 28 °C. The thermal tuning of the LCP emission with a broad tuning range of 49 nm was achieved within a temperature interval of only 3 °C, whereas the wavelength of the RCP emission was relatively stable. Due to the relative simplicity of fabrication and the flexible and tunable abilities with regard to wavelength, our design provides a new possibility for the fabrication of dual-wavelength lasers with orthogonal circular polarizations, which might play an important role in practical applications in the development of photonic devices, display technology, biological sensors, and quantum information communication.

## Figures and Tables

**Figure 1 polymers-15-01226-f001:**
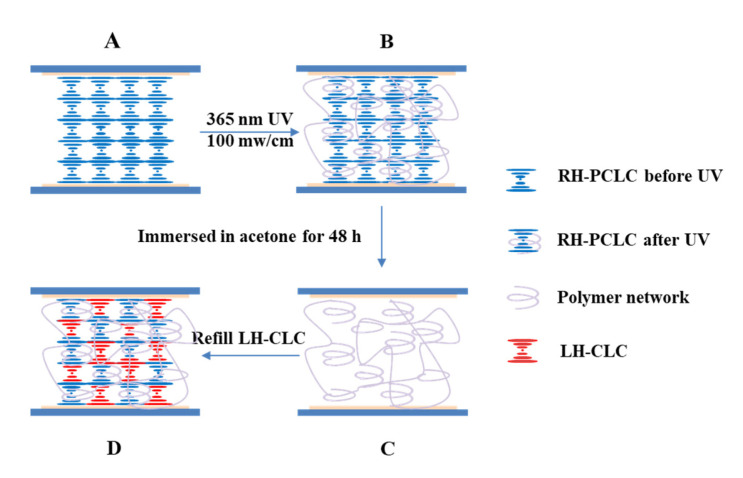
A schematic diagram of the preparation of the superstructure. (**A**) RH-PCLC before UV exposure (Sample A), (**B**) RH-PCLC after UV exposure (Sample B), (**C**) pure polymer network (Sample C), (**D**) polymer network refilled with the LH-CLC (Sample D).

**Figure 2 polymers-15-01226-f002:**
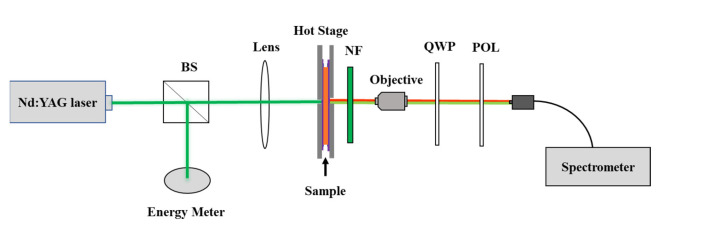
Schematic diagram of the experimental setup: BS: beam splitter; NF: notch filter; QWP: quarter-wave plate; POL: polarizer.

**Figure 3 polymers-15-01226-f003:**
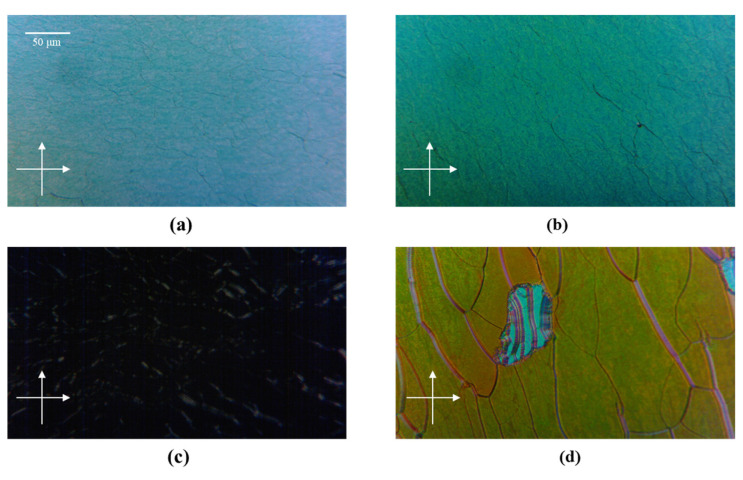
The polarizing optical microscopy images of samples at different preparation phases. (**a**) RH-PCLC before UV exposure (Sample A), (**b**) RH-PCLC after UV exposure (Sample B), (**c**) pure polymer network (Sample C), (**d**) polymer network refilled with the LH-CLC (Sample D).

**Figure 4 polymers-15-01226-f004:**
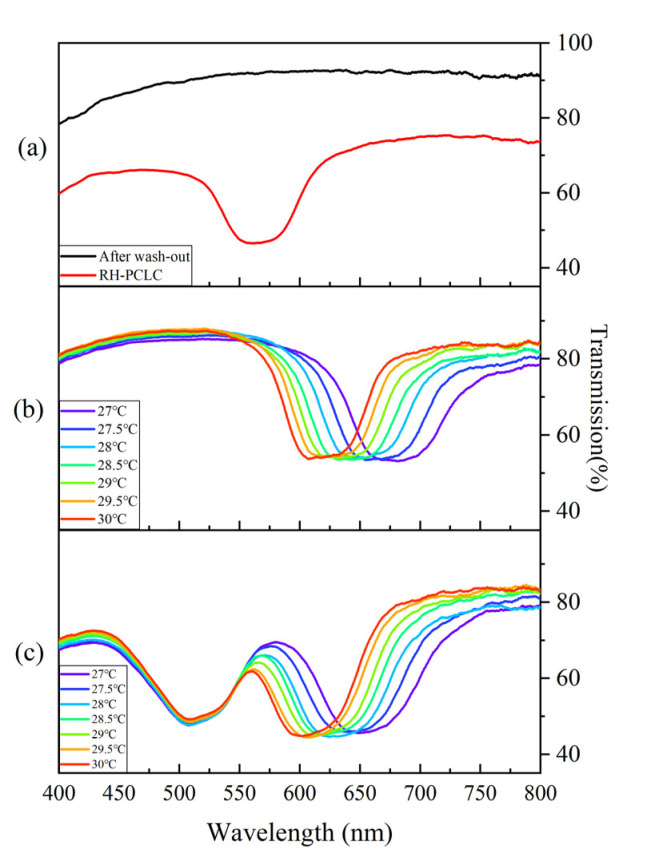
Transmission spectra (**a**) Sample B (red line), Sample C (black line), (**b**) the LH-CLC in a traditional liquid crystal cell (Sample E) at different temperatures, and (**c**) Sample D at different temperatures.

**Figure 5 polymers-15-01226-f005:**
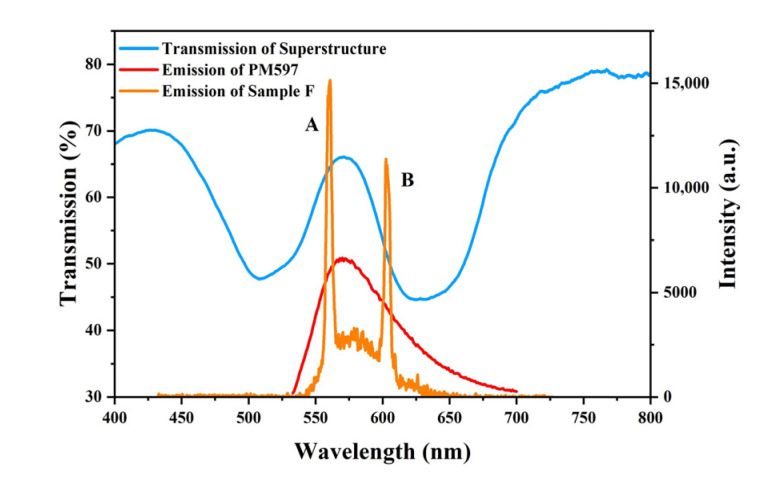
Transmission spectrum (Sample D, blue curve), emission spectrum of the dye measured in the isotropic phase of the LH-CLC material (PM597, red curve), and lasing emission spectrum in the superstructure with dual chirality (Sample F at 28 °C, 1.5 μJ, orange curve).

**Figure 6 polymers-15-01226-f006:**
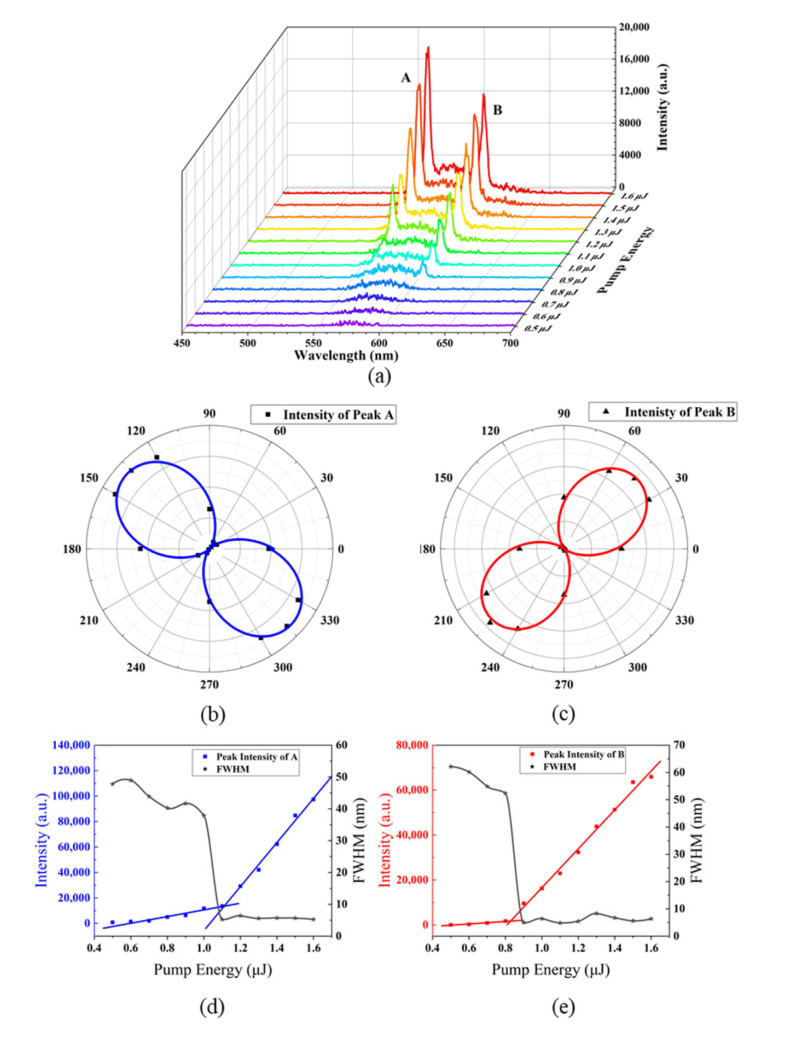
(**a**) Dual-wavelength laser emission spectra of sample F for different pump energies at 28 °C; (**b**,**c**) the polarization states of Peaks A and B, respectively; (**d**,**e**) the dependence of emission intensity and FWHM of Peaks A and B as a function of pump energy, respectively.

**Figure 7 polymers-15-01226-f007:**
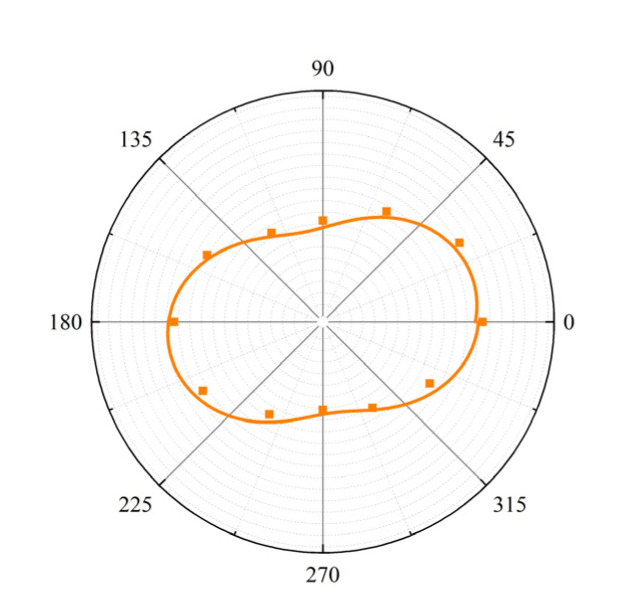
Dichromatic fluorescence spectrum of PM597 dissolved in a planarly aligned LC cell of pure E7 as a function of the angle between the polarizer and the rubbing direction.

**Figure 8 polymers-15-01226-f008:**
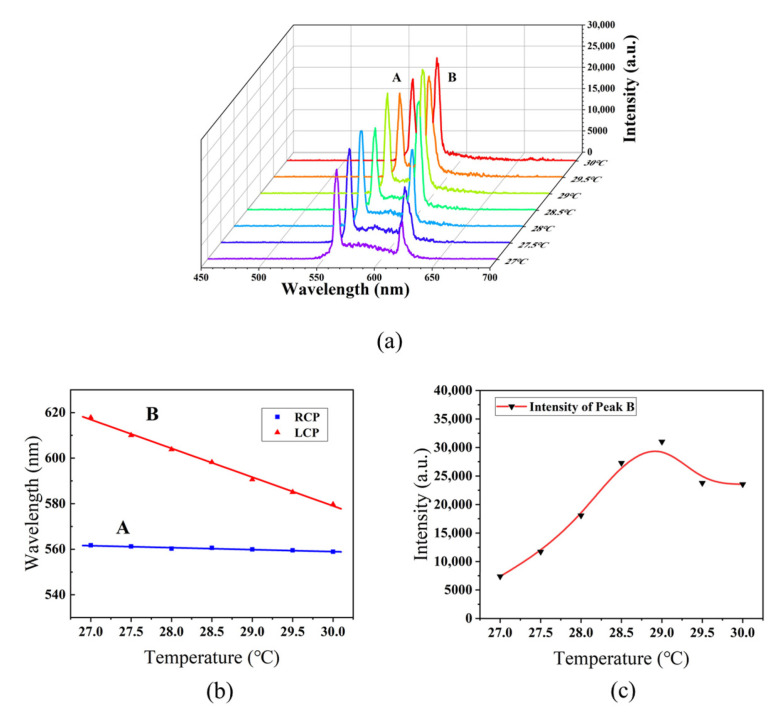
(**a**) Dual-wavelength laser emission spectra at different temperatures; (**b**) lasing wavelengths with orthogonal circular polarizations as a function of temperature; (**c**) lasing intensity of Peak B as a function of temperature.

## Data Availability

The data that support the findings of this study are available from the corresponding author upon reasonable request.

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
