# Peer review of "Dual-Wavelength Lasing with Orthogonal Circular Polarizations Generated in a Single Layer of a Polymer–Cholesteric Liquid Crystal Superstructure"

_polymers, 2023, doi:10.3390/polym15051226_

Round 1

Reviewer 1 Report

The authors report the successful fabrication of a liquid crystal cell containing both right- and left-handed helical structures and left- and right-handed circularly polarized laser emission from the cell. The results are novel and worth publishing. This paper is well organized but could be better by adding detailed information. The following comments should be taken into account and corrected.

1, Polarized light micrographs of the cells should be included. Two selective reflection bands emerged in the optical measurement. This indicates that there are right-handed and left-handed helical segments. In the right-handed helical segments, the liquid crystal, which is spontaneously left-handed, was forced to be right-handed by the polymer helical structures. To force the handedness of the LC, the polymer structure should be dense. On the other hand, the right-handed polymer structure must be absent or thin to a sufficiently large extent in the areas that are left-handed helices. This suggests the cell should be non-uniform in the plane or the thickness direction. Polarized light micrographs will help readers to determine this.

2, It is necessary to clearly show the basis for the structures shown in Fig. 1 (c) and (d). As noted in the comment above, the polymeric structures must be heterogeneous. Fig. 1 (c) and (d) are probably scientifically incorrect.

3, Regarding Figure 2, the names of objects do not match in the caption and in the text. For example, the notch filter in the text is SF, and the analyzer is Pol.

4, Laser oscillation occurs on both the long and short wavelength edges of the selective reflections. It is known that the oscillation on the long wavelength side is stronger when the transition dipole of the dye is parallel to the director, while the oscillation on the short wavelength side is stronger when the transition dipole is perpendicular to the director. The present results suggest that the orientation of the dye in the lc is low. It would be interesting to evaluate the orientation of the dye by measuring the fluorescence dichroism ratio by dissolving the dye in pure E7.

5, The wavelength scales in Fig. 6(b) should be combined into one. The current writing style may give a misleading impression that the laser oscillation wavelengths are interchanged.

Reviewer 2 Report

The article "Dual-wavelength lasing with orthogonal circular polarizations generated in a single layer of a polymer-cholesteric liquid crystal superstructure" by Donghao Yang and co-authors presents the experimental investigation of dual-wavelength orthogonal circular polarizations lasing. Lasing was obtained in the domain structure of a polymer-cholesteric liquid crystal. The right-handed domains are formed by a right-handed polymer scaffold between which the cholesteric LC forms left-handed domains. The dependencies of emission intensity from the pump energy and the lasing wavelengths from temperature are studied. The results obtained are original and can interest to practical applications in photonics.

Research done to a high standard, the manuscript well structured and readable. I recommend the manuscript for publication in Polymers after minor revision.

A few comments on the manuscript are as follows:

1. Since the ratio between the intensities of peaks A and B depends on the pump energy, then pump energy at which the Sample F emission spectrum was taken (orange curve in Figure 4) should be indicated.

2. It is not clear from Figures 4 and 5a how to determine the spectral position and FWHM of emission peaks A and B when the pump energy is less than the threshold one. Additional explanations of the measurement of peaks half-width should be added in the manuscript.

3. Figure 6а shows that the spectral position and intensity of peak B depend on temperature, but the dependence of the peak B intensity on temperature is not explained and discussed. The influence of temperature on the dependence of the emission intensity of peaks from the pump energy should be discussed in the manuscript.
